# SOX2 Is a Univocal Marker for Human Oral Mucosa Epithelium Useful in Post-COMET Patient Characterization

**DOI:** 10.3390/ijms23105785

**Published:** 2022-05-21

**Authors:** Eustachio Attico, Giulia Galaverni, Elisa Bianchi, Lorena Losi, Rossella Manfredini, Alessandro Lambiase, Paolo Rama, Graziella Pellegrini

**Affiliations:** 1Centre for Regenerative Medicine “Stefano Ferrari”, University of Modena and Reggio Emilia, 41125 Modena, Italy; giulia.galaverni@unimore.it (G.G.); elisa.bianchi@unimore.it (E.B.); rossella.manfredini@unimore.it (R.M.); graziella.pellegrini@unimore.it (G.P.); 2Department of Life Sciences, Unit of Pathology, University of Modena and Reggio Emilia, 41124 Modena, Italy; lorena.losi@unimore.it; 3Department of Sense Organs, Sapienza University of Rome, 00169 Rome, Italy; alessandro.lambiase@uniroma1.it; 4Cornea and Ocular Surface Unit, IRCCS San Raffaele Scientific Institute, Vita-Salute San Raffaele University, 20121 Milan, Italy; rama.paolo@hsr.it; 5Holostem Terapie Avanzate s.r.l., 41125 Modena, Italy

**Keywords:** aniridia, biomarker, COMET, cornea, homeostasis, LSCD, ocular surface, oral mucosa, PAX6, SOX2

## Abstract

Total bilateral Limbal Stem Cells Deficiency is a pathologic condition of the ocular surface due to loss or impairment of corneal stem cell function, altering homeostasis of the corneal epithelium. Cultivated Oral Mucosa Epithelial Transplantation (COMET) is the only autologous treatment for this pathology. During the follow-up, a proper characterization of the transplanted oral mucosa on the ocular surface supports understanding the regenerative process. The previously proposed markers for oral mucosa identification (e.g., keratins 3 and 13) are co-expressed by corneal and conjunctival epithelia. Here, we propose a new specific marker to distinguish human oral mucosa from the epithelia of the ocular surface. We compared the transcriptome of holoclones (stem cells) from the human oral mucosa, limbal and conjunctival cultures by microarray assay. High expression of SOX2 identified the oral mucosa vs. cornea and conjunctiva, while PAX6 was highly expressed in corneal and conjunctival epithelia. The transcripts were validated by qPCR, and immunological methods identified the related proteins. Finally, the proposed markers were used to analyze a 10-year follow-up aniridic patient treated by COMET. These findings will support the follow-up analysis of COMET treated patients and help to shed light on the mechanism of corneal repair and regeneration.

## 1. Introduction

The ocular surface is the first passage for the light and acts as an anatomical barrier for the eye against external pathogens. It is comprised of closely integrated regions that form a unique functional unit, comprehensive of different tissues [1,2].

Located between cornea and conjunctiva, the human limbus contains limbal epithelial stem cells of the cornea (LESCs), hosted in pigmented crypts called palisades of Vogt and in other specific crypts [3,4]. Limbal integrity and physiology can be compromised due to primary (genetic) or secondary (acquired) etiologies. The former includes aniridia and other congenital syndromes, while the latter are mainly due to burns and inflammatory/infectious insults [5,6].

If LESCs are damaged, their essential function of continuous regeneration of the corneal epithelium is lost [7], leading to a pathological condition called Limbal Stem Cell Deficiency (LSCD). LSCD (Orpha:171673) is a rare disease with a prevalence of 1–5/10,000 [8] and can be classified as partial if some part of the limbus is healthy or total when the limbus is completely damaged. These two conditions may arise in one (unilateral) or both (bilateral) eyes, with the second condition resulting as the most severe (total bilateral LSCD).

In the case of severe but not total LSCD, the most conservative treatment for both eyes was proposed by Pellegrini and colleagues in 1997, introducing the reconstruction of an ex vivo limbal epithelium starting from a small biopsy of spared autologous limbus [9]. This regenerative medicine approach was registered as an ATMP with the commercial name of Holoclar^®^, currently supplied to patients all over Europe [10].

However, this treatment is not feasible in the case of total LSCD because no residual limbal stem cells are available for ex vivo culture. In this condition, an alternative autologous cell population must be found. Among several cells sources (human embryonic stem cells, oral mucosal epithelial cells, conjunctival epithelial cells, mesenchymal stem cells, induced pluripotent stem cells, and others), perhaps the most suitable source is the oral mucosal epithelium. Indeed, the oral mucosa embodies favorable features such as high regenerative capacity, easy accessibility for biopsy withdrawal, and absence of “keratinization” [11,12], namely the formation of the cornified envelope.

The transplantation of cultured epithelial oral mucosa sheets onto the ocular surface of patients with total bilateral LSCD resulted in well-tolerated and effective mid-term and long-term follow-up analysis [13,14]. This procedure, named Cultivated Oral Mucosa Epithelial Transplantation (COMET), was also reproduced by several groups [15]. 

Two possibilities can explain the positive outcomes obtained in corneal regeneration with COMET: (i) epithelial stem cells within the transplanted oral mucosal sheets might regenerate in vivo the ocular surface (“engraftment” hypothesis), or (ii) residual LESCs might proliferate and regenerate the ocular surface (“stimulatory” hypothesis) [16,17]. The first possibility implies a “total anatomical deficiency” of LECSs, while the second involves a “functional deficiency”, where few otherwise undetectable residual cells need specific paracrine factors from donor cells and an appropriate microenvironment to proliferate and differentiate [18]. In support of this second theory, previous studies on limbal allograft transplantation suggest that some objective and subjective improvements may occur even in the absence of detectable levels of donor stem cells survival onto the ocular surface [19,20].

It is not possible to confirm which biologic mechanism underlies the repair of the human ocular surface after COMET. Indeed, there are no specific markers capable to distinguish unequivocally oral mucosa cells from corneal and conjunctival cells. It means that follow-up analyses performed on samples derived from COMET patients (i.e., impression cytology or penetrating keratoplasty) cannot provide a specific picture of the epithelial type. Hence, finding a differential marker is the primary goal for understanding corneal repair’s biological mechanism.

Cytokeratins are epithelial-specific intermediate filaments expressed in a tissue-specific and differentiation-dependent manner and have been widely used to characterize the epithelia, as mentioned above [21]. In particular, cytokeratin 3 (K3) and cytokeratin 12 (K12) are renowned markers of the corneal and limbal epithelium, while cytokeratin 13 (K13) is the principal keratin of conjunctival epithelium [22]. On the other side, the oral mucosa expresses both K3 and K13 [21]. Phenotypic findings of samples derived from successful COMET transplants performed by numerous groups revealed the expression of K3, K4, and K13 onto the ocular surface, alongside with the absence of K12 and MUC5ac (a marker of conjunctival goblet cells). This pattern was expanded with the negative expression of K8, K19, and PAX6 [23,24].

Despite being proposed by several groups [16,23,24,25], the use of cytokeratins as markers to distinguish among the different epithelia revealed some disadvantages. Firstly, these oral mucosa markers are often in vivo expressed in the two ocular districts (which is the case for K3 and K13). Secondly, some keratins may change their expression in wound healing and pathologic conditions. This is the case of K6, K16, and K17, which are expressed in proliferating but not resting epithelial keratinocytes [26]. The expression switch is probably due to growth factors and cytokines such as IL-1, TNF-α, TGF-β and IFN-γ [27,28]. In addition, the downregulation of the corneal keratin K12 was observed with the overexpression of PAX6, which is a transcriptional factor fundamental for eye development, whose absence is also involved in some forms of LSCD [29].

Our previous studies highlighted the expression of some keratins in vitro that were absent in resting condition in vivo [30]. This activation could be explained by a “wound healing-like” condition of the cultured cells, stimulated to proliferation and migration to reach the confluence as in the case of wound closure.

Altogether, these observations suggest that the keratins alone cannot be considered sufficient and reliable markers to distinguish oral mucosa from the corneal epithelium unequivocally.

In the current study, we decided to gather unbiased data through microarray analysis; this approach enabled the identification of a subset of mainly up- and downregulated markers in the comparison between the different epithelia. Starting from these findings, we validated the expression of the chosen transcripts at the protein level and tested them to determine their presence (or absence) in a specimen from a patient previously treated with COMET.

This work will support the characterization of the epithelium of patients after COMET, thus understanding the biologic mechanism that drives corneal repair.

## 2. Results

### 2.1. Keratins Expression in Wound Healing

Keratins have been widely used to investigate the presence of oral mucosal tissue in COMET-treated ocular surfaces [24]. The keratin 3 (K3) and keratin 13 (K13) have been the most employed markers, the former being expressed in central cornea, limbus and oral mucosa, the latter in conjunctiva and oral mucosa [22].

We investigated K3 and K13 expression in vivo and in vitro. As shown in Figure 1A, K3 was detected in oral mucosa and limbus/cornea both in vivo and in vitro, while it was occasionally expressed in cultured conjunctival colonies, despite its absence in in vivo staining. As expected, K13 was only detected in in vivo samples of oral mucosa and conjunctiva, whereas in vitro it was present only in large and differentiated cells of cultured limbo-corneal colonies. Indeed, the expression of K13 recapitulates what happens in the wound healing, which is the condition of the post-transplanted cornea.

Due to the discrepancy between in vivo and in vitro expression, these two keratins cannot be reliable markers to identify the oral mucosal epithelium, specifically in a pathologic environment.

### 2.2. Gene Expression Profiling of Oral Mucosa, Conjunctiva and Limbal Holoclones

Stem-cell-enriched holoclones derived from three in vitro cultured tissues were iso-lated by clonal analysis (n = 2 donors for each epithelium) as previously described [31], amplified and processed for RNA extraction. Collectively, 32 holoclones (n = 9 from conjunctiva, n = 8 from limbus and n = 15 from oral mucosa, Appendix A) were profiled by microarray.

The principal component analysis (PCA) on pre-processed, unsupervised data and the hierarchical clustering based on the list of differentially expressed genes in each pairwise comparison show that holoclones derived from the same epithelium clustered together. Besides, limbal holoclones were transcriptionally located between the oral mucosa and the conjunctival holoclones (Figure 1B–D). Indeed, the ANOVA-based pairwise comparison among the three epithelia highlighted a remarkably higher number of differentially expressed genes (DEGs, fold change FC ≥ 2 and FDR < 0.05 for upregulated; FC ≤ −2 and FDR < 0.05 for downregulated) in oral mucosa vs. conjunctiva with respect to the other pairwise comparisons (Figure 1C,D, Spreadsheets S1–S3).

### 2.3. Markers for Oral Mucosa Compared to Ocular Surface Tissues

To identify novel tissue-specific markers, we focused on the most differentially expressed transcripts in the pairwise comparisons between the oral mucosa and the ocular epithelia (Appendix A). The most upregulated transcript in the ocular surface epithelia compared to oral mucosa was PAX6 (FC = 118.41 and FC = 195.81 in limbus vs. oral mucosa and limbus vs. conjunctiva, respectively). This transcriptional factor was already well-known as the master regulator of eye development [31] (Figure 2A).

On the other hand, one of the most upregulated transcripts in the oral mucosa compared to the ocular surface epithelia was SOX2 (FC = 30.32 and FC = 33.10 in oral mucosa vs. limbus and conjunctiva, respectively) (Figure 2A).

### 2.4. Validation of the Results by Real-Time RT-PCR

Real-time RT-PCR analyses confirmed that Pax6 and Sox2 were respectively up- and downregulated in the holoclones of the corresponding tissues (Figure 2B). Moreover, Pax6 mRNA was absent in holoclones, meroclones and paraclones derived from oral mucosa (Appendix A), as well as in low, medium and old passages of the oral mucosa strains cultured until replicative senescence (Appendix A).

### 2.5. Validation of the Results by Immunofluorescence, Immunohistochemistry and Western Blot

To evaluate these two putative tissue markers at the protein level, we performed indirect immunofluorescence staining on OCT frozen samples of in vivo limbo-cornea and oral mucosa samples. PAX6 was completely absent throughout the oral mucosa, while it was expressed in many nuclei of all layers in the central cornea and limbus (Figure 3A). Concerning SOX2 protein, we found a nuclear expression in basal and suprabasal tiers of oral mucosa epithelium, whereas we could not detect any expression in the central cornea, limbus, and conjunctiva (Figure 3A).

Both markers were also validated by immunohistochemical analysis on FFPE (Figure 3B), and results were comparable with immunofluorescence data. Besides, their expression was analyzed in the in vitro wound healing model by immunofluorescence analysis on cultured cells at young passages. The expression pattern did not change compared to in vivo staining; PAX6 was still present in many nuclei of limbal and conjunctival but not in oral mucosal cultures (Figure 3C). On the contrary, SOX2 was present in some oral mucosa nuclei and absent in all limbal and conjunctiva cultured cells (Figure 3C).

In vitro protein expression of PAX6 and SOX2 was also evaluated by Western blot analysis. Lysates derived from early passages of the cultured limbus, conjunctiva and oral mucosa (n = 3) were analyzed (Figure 3D). Bands corresponding to PAX6 and SOX2 appeared only in ocular surface and oral mucosa lysates, respectively. 

Results suggest that the in vivo and in vitro patterns of PAX6 and SOX2 transcriptional factors are reliable to distinguish the oral mucosa from ocular tissues. Notably, in contrast with keratins, all the tested conditions highlighted no expression change of these markers. 

### 2.6. SOX2 Is Negatively Correlated to Oral Mucosa Stratification

In oral mucosa epithelium, SOX2 has been correlated to an increased cell migration and improved wound resolution in vivo [32].

To evaluate the in vitro SOX2 putative role in oral mucosa epithelium, three primary oral mucosa strains were cultured up to one week after confluence, promoting stratification and differentiation. Then, we analyzed the expression of several markers (including SOX2) by Western blot in both low and induced multilayer differentiation conditions. After stratification of the epithelial cultures, SOX2 was significantly downregulated, similarly to p63 (alpha isoform), a well-known marker of epithelial proliferative potential (Figure 3E). On the contrary, K13 and involucrin, strongly related to differentiation [33], increased in stratified epithelia. Nonconfluent and stratified cultures of limbal keratinocytes were included as control, showing a consistent decrease in p63 alpha expression in the latter condition. K13 was detected in limbal cultures, as previously demonstrated by immunofluorescence (Figure 1A), due to a culture condition that mimics a wound healing state.

Along with the in vivo SOX2 staining in basal and suprabasal cell layers (Figure 3A,B), the results highlighted that this transcriptional factor is inversely related to oral mucosa epithelial differentiation. Considering these findings, putative roles concerning its ‘stemness’ or proliferative functions still need to be explored.

### 2.7. Phenotypic Characterization of Patient after COMET 

To assess the usefulness of SOX2 for identifying oral epithelium on the cornea after transplantation, we analyzed the corneal buttons obtained after full-thickness corneal transplantation (PK) from a female patient with aniridia who previously underwent COMET.

Before the surgery, the cornea of the left eye was entirely covered with fibrovascular pannus, with blood vessels migrating to the central cornea. After COMET treatment, the superficial corneal vessels significantly decreased, the epithelium resulted intact and stable, although the corneal stroma remained opaque. Due to the opacity and further complications, the patient repeated the penetrating keratoplasty (PK) at 1, 3, and 10 years from COMET surgery. The removed corneal buttons were subjected to histological and immunohistochemical analysis (Figure 4A).

A defined SOX2 positivity was present in the nuclei of the basal layer of several samples derived from the patient’s corneal buttons of all the PKs (Figure 4A). Notably, SOX2 positive areas were found in the central zone of corneal button samples collected 1- and 3-year post-COMET, while 10 years later only a few spared marginal areas of SOX2 positive cells were detected. We also examined the presence of the K13 differentiation marker; a strong positivity was noted across all the samples, particularly in the suprabasal tiers of the stratified areas (Figure 4A,B). Indeed, the histological characterization revealed areas with a marked stratification (up to 10–12 layers of cells) compared to other less stratified corneal-like zones (about 3–5 layers of cells). Together with SOX2 and K13 positivity, the increased stratification suggests the presence of areas of oral mucosal epithelium onto the patient’s transplanted eye up to 10 years after COMET. In contrast to SOX2, PAX6 positive regions increased among the three time-points, reaching an average of 73.2% positive areas after 10 years (Figure 4A,B).

Furthermore, to investigate the presence of conjunctival epithelium within the specimens, the Alcian Blue/PAS staining was performed to identify goblet cells (acid type mucins are outlined in blue). As shown in Figure 4, the most stratified areas of the corneal button were devoid of goblet cells, while several less stratified areas showed the presence of clustered conjunctival goblet cells. Finally, none of the samples revealed the expression of corneal keratins K3 and K12, suggesting that PAX6 positive regions could be associated with the presence of conjunctiva devoid of goblet cells due to an altered environment rather than corneal epithelium (Figure 4A,B).

To summarize, within the 10-years post-COMET, the ocular surface of this patient suffering from aniridia revealed the coexistence of areas comprising oral mucosa tissue (SOX2+/K13+) and conjunctival epithelium (presence of goblet cells and K13+). 

## 3. Discussion

The oral mucosa epithelium proved to be a promising source of cells to treat bilateral LSCD, with COMET standing as an optimal autologous tissue-engineered approach to overcome the many limitations related to allogenic transplants and donor site morbidity, among others [18,34].

In this scenario, understanding the biological process that drives corneal repair after cultured autologous oral mucosa transplantation is relevant for conducting an accurate patient selection in the future clinical trial, as well as for tackling clinical decisions regarding post-COMET treatments and the best way to conduct follow-up analysis.

The current hypothesis concerning the mechanism of action are (i) engraftment of oral mucosa graft on patients’ ocular surface or (ii) biological stimulation of few residual corneal cells. While the former event can be inferred in the case of “anatomical LSCs deficiency”, with oral mucosa or conjunctiva being the only two epithelia available for repair, the latter can be proven by the evidence that recipient corneal cells are found in the long- term follow-up over LSCD transplanted eyes [19,20,35]. Indeed, it has been thought that engrafted donor cells could provide a paracrine stimulation of a few recipients’ residual autologous corneal cells, hampered by pathological conditions such as inflammation and the presence of conjunctival vessels. Under this altered environment, the few residual autologous corneal cells could not be able to regenerate the ocular surface, resulting in a “functional deficiency” of limbal stem cells [17]. 

An additional explanation for this phenomenon could be the transdifferentiation (also called “plasticity”) of oral mucosa into corneal cells under different environmental stimuli [36]. However, this hypothesis does not appear consistent with previous observations, showing that adult epithelial cells retain their differentiation program even after engraftment in other human bodies’ epithelial sites [35,37]. 

Cytokeratins have been widely used to distinguish the type(s) of tissue detectable onto the transplanted ocular surface due to their high expression in epithelial tissues, as well as for their structural role as intermediated filaments [18,21]. The ocular surface has been extensively characterized by cytokeratin expression [21]. K3 and K12 have been associated with the corneal tissue, while K7, K13 and K19 were mostly related to the conjunctiva. During differentiation, oral mucosa expresses K3, K4 and K13 [38]. However, these proteins cannot be used as unique biomarkers for oral mucosa, because they are also expressed in the cornea (K3 and K4) or conjunctiva (K4 and K13) [23].

In wound healing, as well as in many pathologic conditions, epithelial cells can be “activated” (i.e., triggered to proliferate), producing proteins different from those expressed during normal homeostasis [28]. In fact, molecules such as growth factors, chemokines and cytokines secreted by surrounding cells can trigger protein synthesis, which is the case of some specific keratins. For instance, K6 and K16, constitutively expressed in hyperproliferative epithelia as mucosae [21], are also markers of the active proliferative condition. The same is for K17, which is involved in intracellular signaling pathways [28].

In past research, Sugiyama and colleagues proposed K6 as a specific marker for oral mucosa in a transplanted rabbit model of LSCD [16]. In their work, one out of three animals displayed a mixed pattern of proteins on the transplanted ocular surface, showing the co-expression of PAX6, K12, MUC5 and K6. However, due to its relationship with proliferation and pathological conditions [28], K6 is an ambiguous marker, and it is highly expressed in activated cornea and conjunctiva specimens (Unpublished data).

In this paper, we propose SOX2—a key transcription factor involved in pluripotency, embryogenesis, fetal development, and wound healing processes [33,39,40]—as the biomarker for discriminating cultivated oral mucosa graft, limbo-corneal and conjunctival epithelia on transplanted eyes of COMET treated patients. In 2019, SOX2 has been identified in mouse adult cornea and barely in human corneal cells cultured in low calcium medium without serum on mitomycin-treated feeder layer [41]. This finding can be explained with the different species (mouse vs. human) and with different culture conditions compared to the clinical-grade system adopted to validate the microarray data of this work.

SOX2 will be helpful to understand how the COMET works in vivo after surgery. Our analyses at 1-, 3-, and 10-years of follow-up showed SOX2 positive areas, suggesting the presence of an engrafted oral mucosa epithelium, correlated with the positive clinical outcome of the transplant. On the contrary, a SOX2 negative ocular surface may indicate either a failure of the transplant (in case of neovascularization and development of a new conjunctival pannus) or the regeneration of a new corneal epithelium after the stimulation of a few spared residual limbal stem cells. Remarkably, SOX2 positive areas were detected in the central corneal button when considering samples at 1- and 3- years of follow-up, while only a few marginal areas were SOX2 positive in the PKs performed 10 years later. This could be explained through a gradual loss of oral mucosal cells.

In this context, another scenario could be considered: a mixed pattern of engraftment and stimulation. Indeed, the histologies of the patient analyzed in this work revealed SOX2/K13 positive areas, but also an expression of PAX6 (ocular surface marker) and the presence of clusters of conjunctival goblet cells, which could indicate partial migration of the conjunctiva over the corneal surface post-transplant. However, the presence of corneal cells is still under debate because PAX6 expression is not matched with the presence of K3 and K12. The negativity for these two corneal markers could be explained either by the absence of corneal cells within the transplanted ocular surface or by the pathologic condition influencing their expression, as previously observed in aniridia and other conditions [39,42]. Moreover, the described inconsistency highlighted the unreliability of the standard molecular markers for distinguishing the different epithelia.

SOX2 is expressed only in the basal layer of the oral mucosa, and it does not identify the differentiated and stratified cells of the upper superficial layers, limiting its detection by impression cytology. 

Future perspectives will include extending this analysis to other markers and extending the study to a larger cohort of patients. Investigations on corneal buttons from different pathologies of oral mucosa transplanted patients will provide more information about the engraftment/stimulation scenarios, and possibly correlate with the patients’ original clinical severity, age, etiologies, etc.

Moreover, the associated analysis by in vivo confocal microscopy (IVCM) could support the examination of the ocular surface of transplanted patients and provide new insights about the microenvironment in case of presence/absence of oral mucosa epithelium [40]. Indeed, in situ oral mucosa has been widely characterized by IVCM [43], and this morphology information could be compared to the grafted oral mucosa, corneal and conjunctival parameters [5,44,45]. Together with biomarkers investigation, this non-invasive methodology will provide a deep, detailed picture of the patient’s ocular surface. 

Finally, understanding if oral mucosa may stimulate any residual limbal stem cells to regenerate an autologous cornea would be crucial for understanding tissue homeostasis and regeneration. Cytokines and extracellular vesicles (EVs), such as exosomes, are released factors primarily responsible for stimulating nearby cells, and recent publications highlighted the high regenerative potential of oral keratinocytes-derived exosomes [46,47]. We can speculate that such secreted factors could be used in combined therapy with cell grafts or as standalone therapeutic agents in case of not-severe conditions of ocular surface damages.

The research of a specific oral mucosa epithelial marker was performed and validated by microarray analysis of holoclones derived from cornea, conjunctiva and oral mucosa tissues. The transcriptomes’ results identified SOX2 and PAX6 as positive and negative markers to assess the presence (or absence) of oral epithelium onto the ocular surface. These findings have been exploited to evaluate COMET patient’s corneal buttons collected after 1-, 3-, and 10-years post-transplantation. These specimens highlighted a mixed pattern, suggesting both the engraftment of the oral mucosa sheet and the presence of ocular cells, mainly derived from the adjacent conjunctiva. The hypothesis of limbal epithelial cells’ stimulation remains a pending issue to be addressed.

## 4. Materials and Methods

### 4.1. Patients and Specimens

Specimens were obtained in accordance with the tenets of the Declaration of Helsinki; donors provided informed consent for biopsies. Permission was also obtained for samples taken from organ donors. Small oral mucosal biopsies were picked up from the inner cheek or inferior labial of patients undergoing surgery for urethral stricture treatment with oral mucosa patch, while corneal and conjunctival specimens were obtained from the ocular surface of donors or cadavers.

### 4.2. COMET Transplantation

A 41-year-old woman suffering from bilateral congenital aniridia with total LSCD and glaucoma underwent an anterior lamellar keratoplasty in her left eye eleven years before the COMET procedure. The lamellar graft failed due to superficial neovascularization caused by limbal deficiency. Visual acuity was hand motion in both eyes.

Under para/retrobulbar anesthesia, the conjunctiva was freed to expose the sclera a few millimeters outside the limbus. Removal of the fibrovascular corneal pannus was then performed. Once the ocular surface was prepared, the oral mucosa cultured on fibrin was transferred on the corneal area, in contact with the appropriate side of the sheet. The excess of the fibrin sheet was trimmed, and the edge was covered with the conjunctiva applying 2 or 3 stitches of vicryl or silk 8/0.

### 4.3. Cell Cultures, Clonal Analysis and Colony Forming Efficiency

Oral mucosal, conjunctival and limbal keratinocytes were obtained from biopsies and cultivated on feeder layer (FL) of lethally irradiated 3T3-J2 cells (a gift from Prof. Howard Green) as previously described [30]. Briefly, cells extracted from biopsy were seeded at a cell density 3–4.5 × 10^4^/cm^2^ on lethally irradiated FL previously plated at the same cell density. Subconfluent primary cultures were then passaged at a density of 6–8.3 × 10^3^ cells/cm^2^. In serial propagation assays, cells were passaged before confluence until replicative senescence. MCF7 and UMSCC14c cell lines were cultured in Dulbecco’s modified Eagle’s medium (DMEM) supplemented with 10% fetal calf serum (FCS) and penicillin/streptomycin. Clonal analysis and Colony Forming Efficiency were performed as previously described [48] and in Appendix A.

### 4.4. Microarray Analyses and Real-Time RT-PCR

Transcriptome analysis was performed by using the Affymetrix HG-133 plus 2.0 array. Keratinocytes subcultured from each holoclone were feeder-depleted using immunomagnetic beads (Miltenyi Biotec, Bergisch Gladbach, Germany). According to the manufacturer’s protocol, total RNA was isolated with the Invitrogen™ PureLink™ RNA Micro Scale Kit (Thermo Fisher Scientific, Waltham, MA, USA). Differentially expressed genes (DEGs) were identified on a robust multiarray average (RMA)-normalized data through the ANOVA module supplied by the Partek GS. 6.6 Software Package [49]. The probesets displaying a fold change contrast ≥ 2 and a false discovery rate (FDR) < 0.05 were selected as DEGs among oral mucosa, limbal and conjunctival holoclones. Integral gene expression data were submitted to the Gene Expression Omnibus repository (http://www.ncbi.nlm.nih.gov/geo; series GSE198408). Real-time RT-PCR was performed to validate microarray data as described within the Appendix A.

### 4.5. Immunofluorescence and Immunohistochemistry

For immunofluorescence (IF) analysis, cells cultured over glass coverslips and OCT-embedded fresh-frozen human tissues were fixed with methanol or 3% PFA for 10 min at −20 °C or room temperature (RT), respectively. Then, samples were permeabilized with 0.2% Triton-X100 in PBS (20 min, RT), blocked with BSA 2% or BSA 2%-FBS 5%-Triton 0.1% (60 min, RT), incubated overnight at 4 °C with the primary antibodies described in Appendix A and thereafter with the appropriate secondary antibodies (60 min, RT). The nuclei were labeled with DAPI (3 min, RT), and slides were mounted with Fluorescent Mounting Medium (Dako, Agilent Technologies, Santa Clara, CA, USA). For immunohistochemical (IHC) analysis, human tissues from biopsies were formalin-fixed paraffin-embedded (FFPE) and sectioned at 3–4 µm. Staining with the respective antibodies (see Appendix A) were run with the Discovery ULTRA Automatic Staining System (Ventana, Roche, Basel, Switzerland) with diaminobenzidine as chromogen and the View DAB Detection Kit (Ventana, Roche, Basel, Switzerland). Antigen retrieval was obtained through the CC1 antigen retrieval buffer (Ventana, Roche, Basel, Switzerland). At the end of the reaction, slides were counterstained with Haematoxylin II (Ventana, Roche, Basel, Switzerland). Alcian Blue-PAS staining was performed using BenchMark Special Stains (Ventana, Roche, Basel, Switzerland). Immunohistochemical sections were acquired using Imager.M2 microscope (Zeiss, Oberkochen, Germany) and AxioVision SE64 software (Zeiss, Oberkochen, Germany). Measure of positive epithelium length for each marker was performed using MosaiX and Length software tools. Average and standard deviation (N = 3) of each marker at each follow-up timepoint was calculated. For statistical analysis 2way ANOVA test was performed using PRISM 8 software (version 8.4.0, GraphPad Software, San Diego, CA, USA).

### 4.6. Western Blot Analyses

Western blot analysis was performed as previously described [48]. Adopted antibodies and products are detailed in Appendix A.

## Figures and Tables

**Figure 1 ijms-23-05785-f001:**
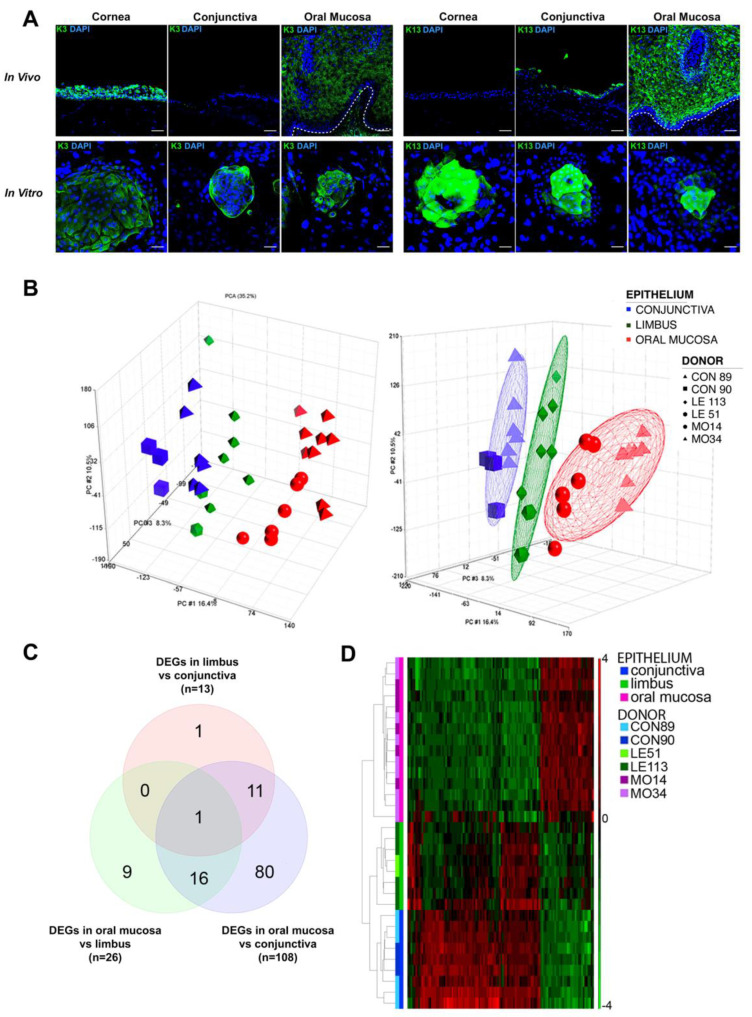
Keratins expression and holoclones microarray analysis. (**A**) Differential expression of K3 and K13 keratins in vivo versus the in vitro condition in cornea, conjunctiva and oral mucosa samples (n = 3). K3 (left side) and K13 (right side) staining in the relevant panels are represented in green, while nuclear staining with DAPI is represented in blue; scale bars = 50 μm; (**B**) Principal component analysis (PCA) of conjunctival, limbal and oral mucosa holoclones. The blue, green and red colors represent conjunctiva, limbal and oral mucosa samples respectively, while each symbol represents a different donor; (**C**) Venn diagram showing the representation of DEGs in each pairwise comparison; (**D**) Heatmap describing gene expression profiles of conjunctival, limbal and oral mucosa holoclones. Red and green represent upregulated and downregulated genes, respectively. Abbreviations: K, keratin; CON, conjunctiva; LE, limbus; MO, oral mucosa; DEGs, differentially expressed genes.

**Figure 2 ijms-23-05785-f002:**
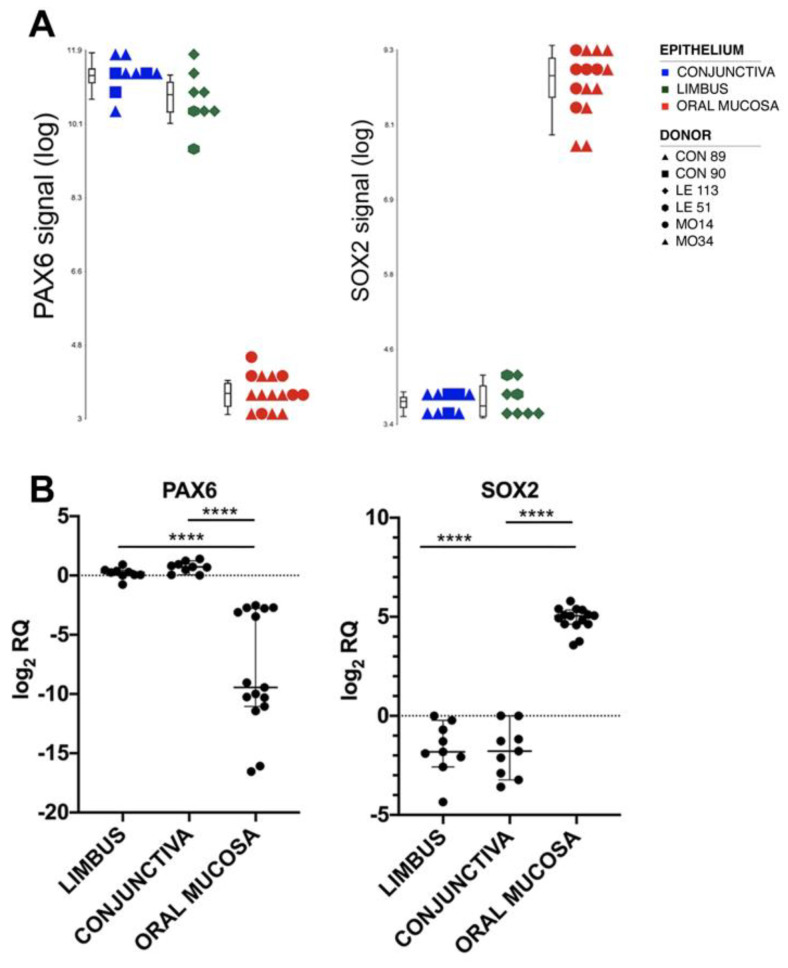
Identification and validation of Pax6 and Sox2 as differentially expressed transcripts. (**A**) Boxplots showing Pax6 and Sox2 signals in conjunctival, limbal and oral mucosal holoclones in log scale; (**B**) Real-time RT-PCR validation of Pax6 and Sox2 transcripts levels expressed as log_2_RQ. **** = *p* < 0.0001 with Mann–Whitney test. Abbreviations: CON, conjunctiva; LE, limbus; MO, oral mucosa; DEGs, differentially expressed genes.

**Figure 3 ijms-23-05785-f003:**
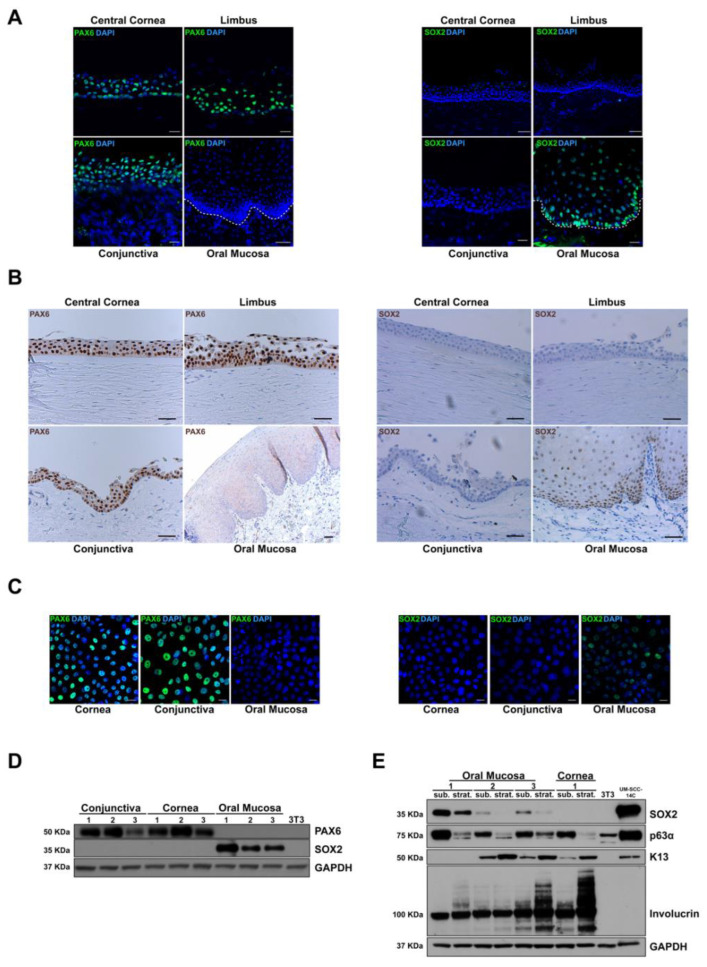
Immunological analysis of PAX6 and SOX2 confirm microarray results also at protein level. (**A**) In vivo immunofluorescence analysis of central cornea, limbus, conjunctiva, and oral mucosa (n = 3). Dotted line marks epithelial basal lamina. PAX6 and SOX2 staining in the relevant panels are represented in green, while nuclear staining with DAPI is represented in blue; scale bars = 20 μm; (**B**) Immunohistochemical analysis on formalin-fixed paraffin-embedded biopsies of central cornea, limbus, conjunctiva, and oral mucosa (n = 3); scale bars = 50 μm; (**C**) Immunofluorescence analysis of young passages of cultured corneal, conjunctival, and oral mucosal cells (n = 3). PAX6 and SOX2 staining in the relevant panels are represented in green, while nuclear staining with DAPI is represented in blue; scale bars = 20 μm; (**D**) Expression of PAX6 and SOX2 detected through Western blot analysis in young passages of conjunctival, corneal and oral mucosa lysates (n = 3). 3T3 cell line was analyzed because used as feeder layer; (**E**) In oral mucosa, SOX2 and p 63 (alpha isoform) were inversely correlated with oral mucosa stratification, highlighted by the increase of K13 and involucrin. 3T3 cell line was analyzed because used as feeder layer; UM-SCC-14c cell line was used as positive control. Abbreviations: sub., subconfluence; strat., stratification; K, keratin.

**Figure 4 ijms-23-05785-f004:**
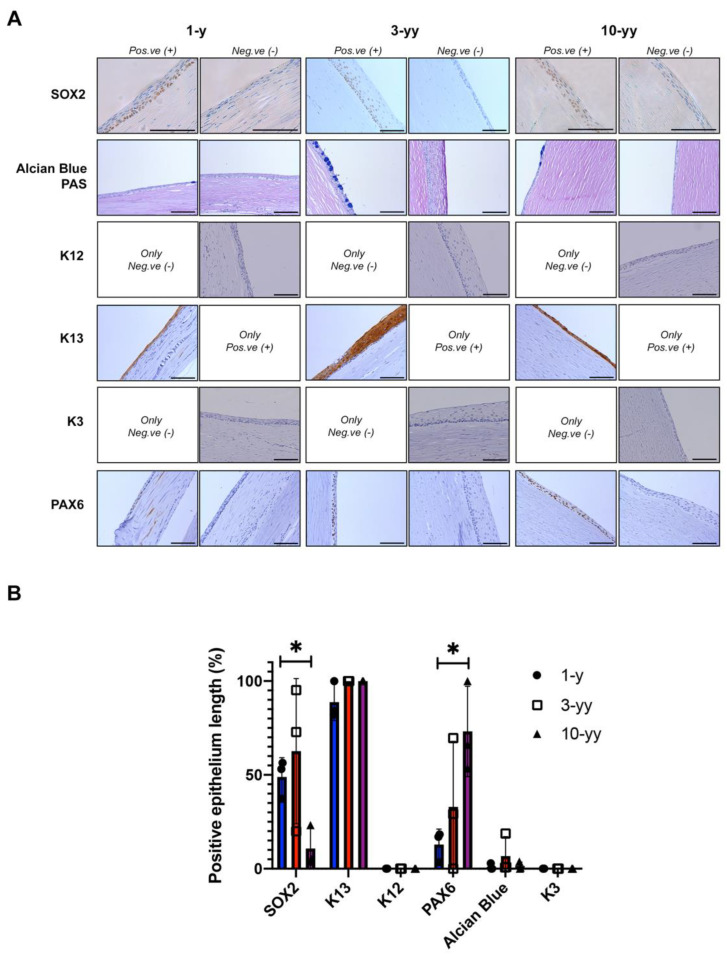
Phenotypic characterization of the corneal buttons derived from the PKs of a patient treated by COMET. (**A**) Immunohistochemical analysis of the corneal buttons from the PKs at 1-, 3- and 10-years post-COMET. Representative images of positive and negative areas are shown in the panel; (**B**) Graph showing the percentage of the positive epithelial length for each analyzed marker. * = *p* < 0.05. Scale bars = 100 μm. Abbreviations: PK., penetrating keratoplasty; y./yy., year/s; pos.ve, positive; neg.ve, negative; K, keratin.

## Data Availability

Integral gene expression data were submitted to the Gene Expression Omnibus repository [48].

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
