# Peer review of "SOX2 Is a Univocal Marker for Human Oral Mucosa Epithelium Useful in Post-COMET Patient Characterization"

_ijms, 2022, doi:10.3390/ijms23105785_

Round 1
Reviewer 1 Report
In this manuscript, Attico et al. have studied biomarkers to differentiate oral mucosa and anterior eye tissues viz. cornea and conjunctiva. They have analysed the DEGs in the transcriptomes of holoclones isolated from tissues of study subjects. Data presented in the manuscript justifies that PAX6 as well as SOX2 can be used as specific markers to distinguish the transplanted oral mucosa in COMET treated patients. Overall study design and presentation of the data in the manuscript is good and was pleasant to read. The reviewer has following two minor suggestions.
Embedded texts in the images are not clearly visible (1A, 3A, 3C). For instance, DAPI in Fig 1A. It is essential to mention the corresponding pseudocolors in the figure legend but not necessary to present as colored embedded texts in the IHC images itself.
Similarly, texts within Fig. 1B are of low resolution and difficult to read.
Author Response
- Embedded texts in the images are not clearly visible (1A, 3A, 3C). For instance, DAPI in Fig 1A. It is essential to mention the corresponding pseudocolors in the figure legend but not necessary to present as colored embedded texts in the IHC images itself.
- Similarly, texts within Fig. 1B are of low resolution and difficult to read.
Reply: We thank the reviewer for his/her interest in our review and for the suggestion regarding visibility of the text within the images, which we think will improve the overall comprehension. As suggested, we have mentioned the corresponding pseudocolors in the figure legend related to images 1A, 3A and 3C, while we have left the colored embedded text in the images changing the color of the word “DAPI” with a brighter one.
We have also replaced Fig. 1B with a higher resolution Figure, where the text is more clearly visible. However, the text within the graphs is not editable therefore we delated one out of three redundant graph and zoom in the other two graphs.
Overall, we hope that these changes have improved the visibility of both the texts and the images.
Please see also the attachment letter to the Editor.

Reviewer 2 Report
The premise of the work was that Cultivated Oral Mucosa Epithelial Transplantation (COMET) is the only autologous treatment for bilateral limbal stem cells deficiency but the markers for oral mucosa identification (e.g., keratins 3 and 13) are also expressed by corneal and conjunctival epithelia, hence, are not specific. The authors proposed SOX2 as a new specific marker to distinguish human oral mucosa from the ocular surface epithelia. Using microarray assay human oral mucosa, limbal and conjunctival cultures, the authors compared the transcriptome of holoclones (stem cells) and identified high expression of SOX2 in oral mucosa vs cornea and conjunctiva, while PAX6 was highly expressed in corneal and conjunctival epithelia. The transcriptomic data were validated by qRT-PCR, as well as at the protein level. The proposed markers were also used to analyze a 10-years follow-up aniridic patient treated by COMET. The authors conclude that their findings will support the follow-up analysis of COMET treated patients. The paper provides interesting and potentially translational data. This reviewer has the following fairly minor concerns about the manuscript.
- Introduction: “The ocular surface is both a barrier and a passage: it is the first anatomical barrier against external pathogens”… Please remove this sentence as misleading. Nasal and oral cavity, and skin should be considered as such as well.
- Introduction: line 70; please add “cornified” before the “envelope”.
- Fig. S1: please explain what is the difference between two graphs in A. Also, it is customary to call passages low, medium, high (also in Fig. 3 legend).
- Line 284: the current term is “penetrating keratoplasty”.
- Fig. 4. Please spell “length” correctly on the Y axis. Is it possible to provide a higher magnification for SOX2 at 1 year and 10 years? Is the presence of K13 and lack of K12 and K3 a sign of conjunctivalization?
- Methods: What was the time between death and processing of samples? It may be critical for the gene and protein expression. It may be assumed that oral mucosal specimens would be “fresher” than the ocular ones.
- Line 469: it should be “Real-time RT-PCR”.
- Please clarify two important fixation-related points. 1. How were the fresh-frozen tissues processed; what was the fixation and for how long. 2. Was any antigen retrieval done on paraffin-embedded tissue?
- The authors’ data contradict the previous publication on mouse cornea about the positive expression of SOX2 in basal epithelial cells. Please cite it and explain. Stem Cells. 2019; 37(3): 417–429.
Author Response
We thank the reviewer for his/her interest in our review and for the helpful and constructive suggestions, which we believe have strongly contributed to improve the clearness of our work.
- Introduction: “The ocular surface is both a barrier and a passage: it is the first anatomical barrier against external pathogens”… Please remove this sentence as misleading. Nasal and oral cavity, and skin should be considered as such as well.
Reply: We agree with the reviewer that the sentence in this way could be misleading; we modified it accordingly.
- Introduction: line 70; please add “cornified” before the “envelope”.
Reply: We thank the reviewer for the suggestion; we have added the word “cornified” within the text.
- S1: please explain what is the difference between two graphs in A. Also, it is customary to call passages low, medium, high (also in Fig. 3 legend).
Reply: The two graphs in S1A represents two separated experiments, where different samples of Holoclones (H, n=2), Meroclones (M, n=3) and Paraclones (P, n=2) derived from the same strain of oral mucosa (MO14) were investigated for their expression of PAX6. For better clarity, we have implemented this explanation within the Figure legend and we added more information in the x-axis.
For better congruency with other articles, we agree to call passages as low, medium and high.
- Line 284: the current term is “penetrating keratoplasty”.
Reply: We agree with the reviewer and we have modified the term accordingly.
- 4. Please spell “length” correctly on the Y axis. Is it possible to provide a higher magnification for SOX2 at 1 year and 10 years? Is the presence of K13 and lack of K12 and K3 a sign of conjunctivalization?
Reply: We thank the reviewer for his/her observation and we apologize for the typo in spelling “length”, which we have corrected in the new version. We also modified Figure 4 by adding images of higher magnification for SOX2 at 1 and 10 years.
In presence of PAX6, the positivity of K13 and lack of K12 and K3 may indicate partial conjunctivalization over the reconstructed ocular surface. However, these k13+ positive cells may not be able to produce goblet cells because of an altered microenvironment. In addition, we commented these results in the discussion at rows 429-435 of the manuscript, as a pathological cornea can envisage altered protein expression (Latta et al., 2018; Auw-Haedrich et al., 2011).
- Methods: What was the time between death and processing of samples? It may be critical for the gene and protein expression. It may be assumed that oral mucosal specimens would be “fresher” than the ocular ones.
Reply: we thank the reviewer for his/her careful observation, which we thoroughly considered in our experiments. Samples derived from oral mucosa and from the patient who underwent COMET were OCT-embedded or formalin-fixed or within 24 hours from the biopsy withdrawal. For the healthy ocular surface samples, the time was more variable, spanning from 1 to 4 weeks. Indeed, these samples derive from Eye Tissue Bank’s leftover (i.e. non-transplantable corneas as a result of infectious or other issues) and are kept in standard preservation medium until fixation.
- Line 469: it should be “Real-time RT-PCR”.
Reply: We agree with the reviewer and we have modified the term accordingly.
- Please clarify two important fixation-related points. 1. How were the fresh-frozen tissues processed; what was the fixation and for how long. 2. Was any antigen retrieval done on paraffin-embedded tissue?
Reply: We thank the reviewer for his/her questions, thanks to which we have now implemented in material and methods section. Fresh-frozen tissues were fixed for 10 minutes with 3% PFA (at Room Temperature) or methanol (at – 20°C), depending on the fixation needed for the corresponding antibody.
Besides, antigen retrieval on paraffin-embedded tissues was obtained through the CC1 antigen retrieval buffer (Roche) used in the automated immunohistochemical analysis.
- The authors’ data contradict the previous publication on mouse cornea about the positive expression of SOX2 in basal epithelial cells. Please cite it and explain. Stem Cells. 2019; 37(3): 417–429.
Reply: We thank the reviewer for citing this article, that we have thoroughly reviewed during our research, and we have now implemented in our manuscript (rows 414-418). In the article by Bhattacharya and coauthors, SOX2 has been identified in mouse adult cornea and barely in cultured human corneal cells. However, several differences can account for these results:
- The human cells were cultured and were not in an in vivo condition;
- The culture system was significantly different from our conditions: low calcium medium without serum and mitomycin-treated feeder layer;
- In fact, we analyzed in vivo and in vitro on humans only, using high calcium medium with serum and lethally irradiated 3T3 as feeder layer.
It is reported that the culture conditions can significantly alter the protein expression and, in our experimental conditions, we did not detect any SOX2 protein expression in adult corneal cells.
Please see also the attached letter to the Editor.
